# Oropouche virus cases identified in Ecuador using an optimised qRT-PCR informed by metagenomic sequencing

Emma L. Wise[1,2], Sully Márquez[3], Jack Mellors[1], Verónica Paz[4], Barry Atkinson[5], Bernardo Gutierrez[6,7], Sonia Zapata[3], Josefina Coloma[8], Oliver G. Pybus[6], Simon K. Jackson[2], Gabriel Trueba[3], Gyorgy Fejer[2], Christopher H. Logue[1,2,3]☯*, Steven T. Pullan[1]☯

1 National Infection Service, Public Health England, Salisbury, United Kingdom, 2 School of Biomedical Sciences, University of Plymouth, Plymouth, United Kingdom, 3 Microbiology Institute, Universidad San Francisco de Quito USFQ, Quito, Ecuador, 4 Hospital Delfina Torres de Concha, Esmeraldas, Ecuador, 5 Arthropod Genetics Group, The Pirbright Institute, Woking, United Kingdom, 6 Department of Zoology, University of Oxford, Oxford, United Kingdom, 7 School of Biological and Environmental Sciences, Universidad San Francisco de Quito USFQ, Quito, Ecuador, 8 School of Public Health, University of California Berkeley School of Public Health, Berkeley, California, United States of America

☯ These authors contributed equally to this work.
* christopher.logue@phe.gov.uk

**Data Availability Statement:** The OROV genome sequences are available from the Genbank database (accession numbers MK506818 - MK506832). The metagenomic sequencing data

## Abstract

Oropouche virus (OROV) is responsible for outbreaks of Oropouche fever in parts of South America. We recently identified and isolated OROV from a febrile Ecuadorian patient, however, a previously published qRT-PCR assay did not detect OROV in the patient sample. A primer mismatch to the Ecuadorian OROV lineage was identified from metagenomic sequencing data. We report the optimisation of an qRT-PCR assay for the Ecuadorian OROV lineage, which subsequently identified a further five cases in a cohort of 196 febrile patients. We isolated OROV via cell culture and developed an algorithmically-designed primer set for whole-genome amplification of the virus. Metagenomic sequencing of the patient samples provided OROV genome coverage ranging from 68–99%. The additional cases formed a single phylogenetic cluster together with the initial case. OROV should be considered as a differential diagnosis for Ecuadorian patients with febrile illness to avoid mis-diagnosis with other circulating pathogens.

## Author summary

Oropouche virus (OROV) causes outbreaks of febrile illness in areas of South and Central America and we recently identified it in Ecuador for the first time, using metagenomic sequencing. The genome sequence data revealed that the Ecuadorian strain of the virus was not detected using a published qRT-PCR, as it differed genetically at the binding site of the reverse primer. To address this, we developed a modified qRT-PCR that showed increased sensitivity for the Ecuadorian strain. This test detected OROV infection in 6 out of 196 febrile patients from Esmeraldas, Ecuador in 2016. OROV was isolated from

are available from the Sequence Read Archive database as fastq files (accession numbers SAMN12241859 - SAMN12241881).

**Funding:** This work was funded by Public Health England as part of a PHE-funded PhD studentship awarded to ELW, SKJ, GF, CHL, and STP. The funders had no role in study design, data collection and analysis, decision to publish, or preparation of the manuscript.

**Competing interests:** The authors have declared that no competing interests exist.

positive patient samples, viral genome sequences were compared to publicly available OROV sequences. This revealed that the Ecuadorian cases are genetically distinct, suggesting that local transmission of the virus should not be ruled out. This work highlights the need for a better understanding of OROV dynamics in Ecuador and surrounding areas, the importance of considering OROV as a cause of fever in Ecuadorian patients and the possibility of selectively using metagenomic sequencing in parallel to traditional molecular techniques in patient testing.

## Introduction

Oropouche virus (OROV) is the causative agent of Oropouche fever, an arboviral illness that is usually self-limiting and mild but in rare cases infects the central nervous system and causes meningitis [1,2]. The virus belongs to the genus *Orthobunyavirus*, family *Peribunyaviridae*, and has a negative-sense, single-stranded RNA genome composed of three segments. The small (S) segment (0.95 kb) encodes the nucleoprotein (N) and non-structural protein (NSs), the medium (M) segment (4.36 kb) encodes a polyprotein consisting of two glycoproteins (Gn, Gc) and one non-structural protein (NSm), whilst the large (L) segment (6.85 kb) encodes the RNA dependent RNA polymerase (RdRP) [3].

OROV is one of the most clinically important orthobunyaviruses in the Americas, with over half a million cases and more than 30 major outbreaks reported since it was first isolated in Trinidad and Tobago in 1955 [4]. These figures are likely to be underestimates, caused in part by underreporting due to the similar clinical presentation of other arboviral diseases that co-circulate in the same areas, including dengue virus (DENV), chikungunya virus (CHIKV), Mayaro virus (MAYV), yellow fever virus (YFV) and Zika virus (ZIKV) [4,5]. The primary vector species implicated in urban OROV transmission is the biting midge *Culicoides paraensis*. This is well supported by both experimental and epidemiological data (reviewed in 5). It has been suggested that the widely distributed mosquito species *Culex quinquefasciatus* may contribute to OROV transmission during outbreaks, although efficiency of transmission has been shown experimentally to be lower than that of the primary vector [1]. Cases of Oropouche fever have been reported in Brazil, Peru, Panama, and Trinidad and Tobago [5]. We recently isolated OROV for the first time from a patient in north-western Ecuador [6] suggesting that OROV may be circulating undetected in this region. This led us to further investigate the presence of OROV in an additional 196 febrile patients from the same coastal area of Esmeraldas, Ecuador, sampled in 2016.

Reverse transcription polymerase chain reaction (RT-PCR) is typically used for the detection of viral RNA directly from clinical samples [7–10]. The rate of detection of OROV RNA in patient plasma during the first five days of illness is reported as 93.3% using a one-step real-time RT-PCR (qRT-PCR) [7]. A disadvantage of RT-PCR is that a conserved target genome sequence is required, making it difficult to detect viral strains that are genetically divergent, reassortant, novel, or of an unexpected virus species. Metagenomic sequencing (untargeted sequencing of all genetic material in a sample) is an attractive alternative to conventional molecular detection methods as it does not require prior knowledge of pathogen genome sequences. For example, an established OROV qRT-PCR [7] was unable to detect OROV in the plasma of an Ecuadorian febrile patient who tested positive using metagenomic sequencing and virus isolation (S segment sequence available at GenBank, accession MF926352.1) [6]. This highlights the potential for misdiagnosis when screening for pathogens that lack a large collection of reference sequences on which to base targeted amplification. Here we report (i)

the development of a qRT-PCR with improved sensitivity, (ii) an algorithmically-designed [16] primer set for whole-genome amplification of OROV, and (iii) the prevalence of OROV in the 2016 Esmeraldas patient cohort.

## Materials and methods

### Patient samples

Plasma samples were obtained from patients at Delfina Torres de Concha Hospital, Esmeraldas province, Ecuador, in 2016 ($n = 196$) and 2017 ($n = 62$), along with metadata: patient age, sex, patient location, number of days of fever, and other clinical signs and symptoms.

### Ethics statement

Blood samples were obtained from Delfina Torres de Concha Hospital (Esmeraldas city, Ecuador), from febrile patients who attended the hospital voluntarily and submitted their blood for routine dengue and Zika virus testing. The use of these samples in this project was approved by the Universidad San Francisco de Quito bioethics committee. Samples were anonymised prior to analysis and the patients were not required to fill out informed consent.

### RNA extraction and PCR assays

Plasma samples were spiked with bacteriophage MS2 ($1.45 \times 10^5$ pfu) as an internal control. RNA was extracted using the QIAamp Viral RNA mini kit (Qiagen) following manufacturer's instructions, substituting the addition of carrier RNA with the same volume of linear polyacrylamide (LPA). A negative extract control (molecular-grade water) was included in each batch of extractions. RNA was screened using Public Health England in-house diagnostic PCR and RT-PCR assays, some of which were adapted from published assays, targeting: DENV [11], CHIKV (in-house), ZIKV [12], YFV [11], MAYV (in-house), *Plasmodium* spp. [13], *Leptospira* spp. (in-house) and *Rickettsia* spp. [14,15].

### Metagenomic sequencing

DNAse treatment, cDNA preparation, random amplification by SISPA (Sequence-Independent Single Primer Amplification), and Illumina sequencing were performed as described previously [17]. Reads were trimmed to remove adaptors and low-quality bases using trimmomatic (0.3.0) [18] with default parameters, to achieve an average phred score of Q30 across the read.

Taxonomic analysis of reads was undertaken using Centrifuge (Galaxy Version 1.0.3-beta) [19] with index "Bacteria, Archaea, Viruses, Human", last updated on 12/06/2016, available at https://ccb.jhu.edu/software/centrifuge/manual.shtml, on a local instance of Galaxy [20]. Mapping to OROV genomes was performed using BWA MEM (v0.7.15) [21], using OROV/EC/Esmeraldas/087/2016 (GenBank accessions MF926352.1, MF926353.1, MF926354.1) as references. Quasibam [22] was used to generate consensus sequences, with a 5x coverage cut off. Mapping statistics were generated using SAMtools (v1.4) [23]. Alignment and analysis of nucleotide and protein sequences was performed using the ClustalW method in MegAlign (v14.0.0). Human reads were removed by mapping to the human genome (human_g1k_v37 [1000 genomes]), non-mapped reads were selected using SAMtools fastq, with flag -F 2. Raw data (stripped of human reads identified via mapping or Centrifuge) is available from the Sequence Read Archive, accession numbers SAMN12241859—SAMN12241881. De novo assembly of non-human reads was performed using SPAdes (v3.8.2) with—meta flag [24]. A BLASTn [25] search was performed on resulting scaffolds larger than 1000 bp, against

GenBank (nr/nt) database. Non-human reads were mapped (BWA MEM) to a multifasta file containing sequences from relevant viruses, including at least one representative genome from each PCR screening target (CHIKV, OROV, DENV1, DENV2, DENV3, DENV4, MAYV, YFV, ZIKV), any virus that had appeared during BLAST analysis of *de novo* assembled scaffolds (Hepatovirus A, Rift Valley Fever virus, influenza A virus), and the internal control MS2.

Samples with >0.02% reads assigned to a virus species by Centrifuge or an assembled scaffold larger than 1500 bp showing homology to a viral sequence using BLASTx were investigated for the presence of that specific virus. A positive virus detection was defined as >40% genome coverage (1x) generated when mapping to a reference genome.

## Virus isolation and quantification

Vero cells, freely provided by the European Collection of Authenticated Cell Cultures (ECACC; catalogue number 84113001), were inoculated with patient plasma diluted 1:10 in Eagle's minimum essential medium (MEM) to a volume of 1 mL, incubated at 37˚C with 5% (v/v) $CO_2$ for 96 hours. Cultures were sampled at 24-hour intervals, virus genome replication was determined using OROV qRT-PCR. Virus was harvested, filtered (0.2 μm pore, Sartorius) and stored at -80˚C.

Plaque assays were performed on Vero cell monolayers. 250 μl of 10-fold virus dilutions were added to cells and incubated for one hour at 37˚C, 5% (v/v) $CO_2$. Inoculum was removed, then 3 mL overlay medium (2x MEM, 1% NEAA, 1% antibiotic-antimycotic, (Invitrogen), 2 mM L-glutamine (Fisher Scientific), 10% FBS (Sigma-Aldrich), and 1% Seaplaque agarose (Lonza)) applied and incubated for four days, before fixation and staining using 20% formalin and 2.3% crystal violet, respectively. Plates were air dried and plaques counted to determine virus titre, using the lowest dilution showing >10 plaques.

## Primer alignment

Primer and probe sequences were aligned using the ClustalW algorithm in the MEGA7 software package, to a set of OROV N gene coding sequences (GenBank) representing the geographic (Brazil, Panama, Peru and Trinidad and Tobago) and temporal diversity of OROV isolates from the 1950s to present day [26].

## qRT-PCR limit of detection analysis

For OROV qRT-PCR assay optimisation (supporting information), a range of primers were tested (S1 Text and S1 Table) using the Superscript III Platinum One-step Quantitative RT-PCR kit (Invitrogen) in a standard 25 μL reaction, run on an ABI 7500 real-time PCR machine (Applied Biosystems). Optimal conditions were 18 μM of each primer, 12.5 μM probe and standard $MgSO_4$ concentration. Cycling conditions: 50˚C for 10 minutes, 95˚C for 2 minutes, 45 cycles of 95˚C for 10 seconds then 60˚C for 40 seconds (with quantification analysis of fluorescence performed at the end of each 60˚C step) and a final cooling step of 40˚C for 30 seconds.

Absolute quantitation of RNA copy number was calculated from a standard curve of target sequence RNA, transcribed using the Megascript kit (Ambion). A 711 bp region of the S segment was amplified from cDNA of OROV/EC/Esmeraldas/087/2016, using primers incorporating a T7 promoter sequence (forward primer sequence 5' GTC AGA GTT CAT TTT CAA CGA TGT ACC ACA ACG G 3', reverse primer sequence 5' GAA ATT AAT ACG ACT CAC TAT AGG G CTC CAC TAT ATG TC 3'). RNA was quantified using the Qubit RNA broad spectrum kit and purity confirmed using the RNA 6000 pico kit on a bioanalyzer (Agilent).

## Multiplex tiling RT-PCR primer design

The S, M and L segment sequences of OROV/EC/Esmeraldas/087/2016 were concatenated and inputted using default settings to the Primal Scheme [16] webtool (http://primal.zibraproject.org/), which implements an algorithm for designing multiplex PCR primers for virus lineages. The designed primer scheme was modified manually to account for segment ends, incorporating 5' and 3' end primers (see supporting information for primer sequences).

RNA from patient sample D-057 and the cell-cultured prototype strain were amplified in two multiplex PCR reactions using the Ecuador OROV primer scheme designed as described above. Primer pools were used at 100 µM. Products were visualised by gel electrophoresis and quantified using Qubit fluorometric quantitation. These were sequenced on a MinION (Oxford Nanopore, UK) and data processed as previously described [16].

## Results

### Development of an OROV qRT-PCR

An existing S segment assay [7] failed to detect OROV RNA in an Ecuadorian clinical sample that tested positive by both metagenomic sequencing and virus isolation (S1 Fig) [6]. The reverse primer (OROV R) [7] had two mismatches to the novel OROV/EC/Esmeraldas/087/2016 sequence. We attempted to improve sensitivity by analysing primer and probe suitability. Two new reverse primers were designed; one with mismatches corrected (Ec R) and one at a new site (Ec2 R) (S1 Table), based on an alignment of all publicly available OROV S segment sequences. Testing was performed against both OROV/EC/Esmeraldas/087/2016 and the 1955 prototype Trinidad and Tobago isolate from a distinct lineage (GenBank KP026181.1). The original reverse primer showed reduced sensitivity for the Ecuadorian strain compared with the prototype strain, which were detected to the $10^{-4}$ and $10^{-6}$ dilutions respectively (Fig 1). Primer Ec R showed improved sensitivity for the Ecuadorian strain but reduced sensitivity for the prototype strain, with detection possible to the $10^{-7}$ and $10^{-3}$ dilutions respectively (Fig 1). Ec2 R improved sensitivity by three logs for the Ecuadorian strain and by one log for the proto-type strain, compared with the original reverse primer (Fig 1). The final assay using Ec2 R underwent validation. The standard curve demonstrated a linear relationship between mean Cq value and log RNA copy number, the absolute limit of detection of the qRT-PCR assay was 10 copies of OROV RNA at a mean Cq value of 38.25 (95% CI 37.8 to 38.7, S2 Fig). No cross-reactivity was observed in exclusivity testing against 23 virus species (S2 Table) within the *Alphavirus*, *Flavivirus*, *Phlebovirus*, *Orthobunyavirus*, *Nairovirus* and *Mammarenavirus* gen-era. Primer and probe sequences were aligned with all publicly available OROV N gene sequences ($n = 149$) to identify mismatches that may reduce sensitivity for certain strains. Of the 149 sequences, 19 had mismatch(es) to the forward primer, 21 sequences showed a mis-match to the probe, and 26 had mismatch(es) to the original reverse primer OROV R, six of which were mismatches at two positions (S3 Table). In contrast, no sequence had more than one mismatch to the new reverse primer Ec2 R, however, the number of sequences with a sin-gle mismatch was higher ($n = 41$).

### Pathogen identification in febrile patients using PCR

We initially used nine PCR or RT-PCR assays to screen serum from Ecuadorian patients with undifferentiated febrile illness for pathogens previously reported from the country. Following our previous metagenomic sequencing based identification of OROV in one of these samples [6], and improvement of the OROV qRT-PCR, we re-screened all patient samples for OROV. PCR screening identified positive cases (Cq value <35) of ZIKV (prevalence 11.2%, $n = 22$),

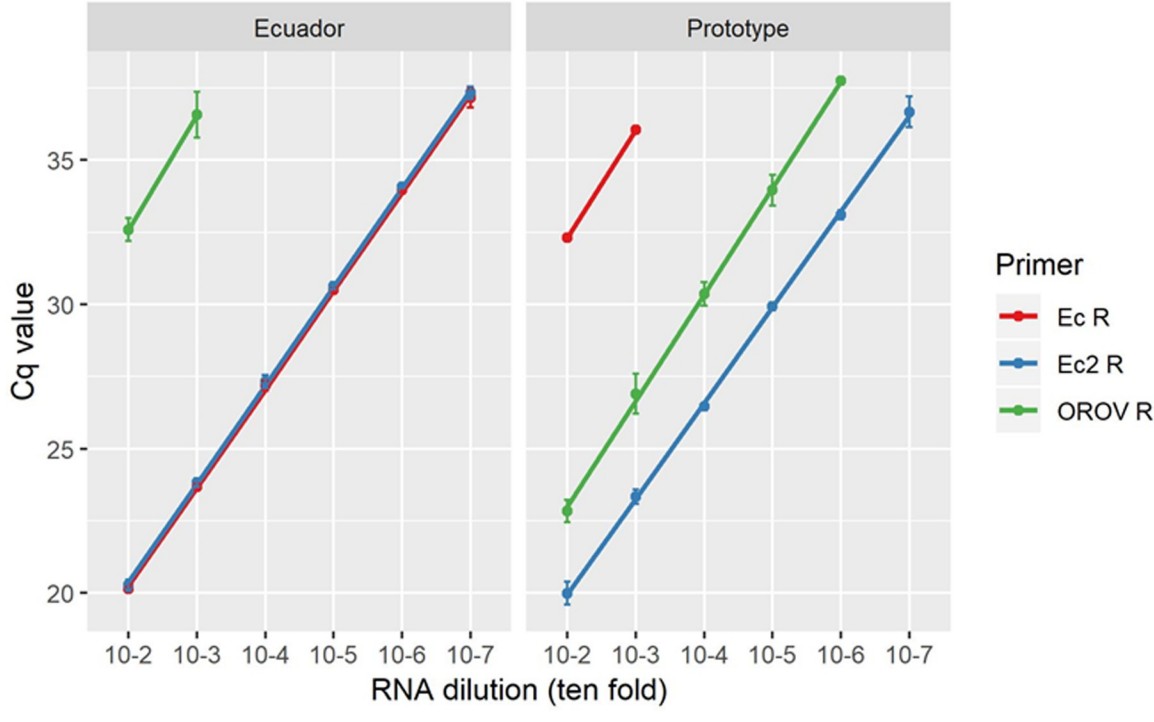

**Fig 1. Sensitivity analysis of three reverse primers used in the qRT-PCR assay.** Cq values are shown for ten-fold serial dilutions of OROV RNA, either from the prototype strain (GenBank KP026181.1), or the Ecuadorian strain OROV/EC/Esmeraldas/087/2016 (GenBank MF926352.1). Three separate experiments were performed, from which the means (plotted as data points) and standard deviations (indicated by error bars) were calculated. Reverse primer sequences (5'–3') are given in the embedded table.

DENV (prevalence 2.0%, $n = 4$), OROV (prevalence 3.1%, $n = 6$), and *Leptospira* spp. (prevalence 0.51%, $n = 1$) (Fig 2). In addition, a number of potentially positive cases (Cq value 35–39.9) were identified (ZIKV $n = 15$, DENV $n = 3$, CHIKV $n = 2$. and *Leptospira* spp. $n = 1$, Fig 2). Cq values in this upper range require additional testing to confirm positive status (this was performed for the Cq >35 OROV sample, but not for the other pathogens). No cases of MAYV, YFV, *Plasmodium* spp. or *Rickettsia* spp. were identified. 147 samples were negative for all pathogens tested. All samples were qRT-PCR-positive for internal control MS2.

## Follow up screening of febrile patient samples from 2017

Sixty-two samples from febrile patients from Esmeraldas, taken in 2017, were tested for OROV, CHIKV, DENV, MAYV, YFV, ZIKV and *Rickettsia* spp. by PCR and RT-PCR. All samples were negative for all pathogens and positive for internal control MS2.

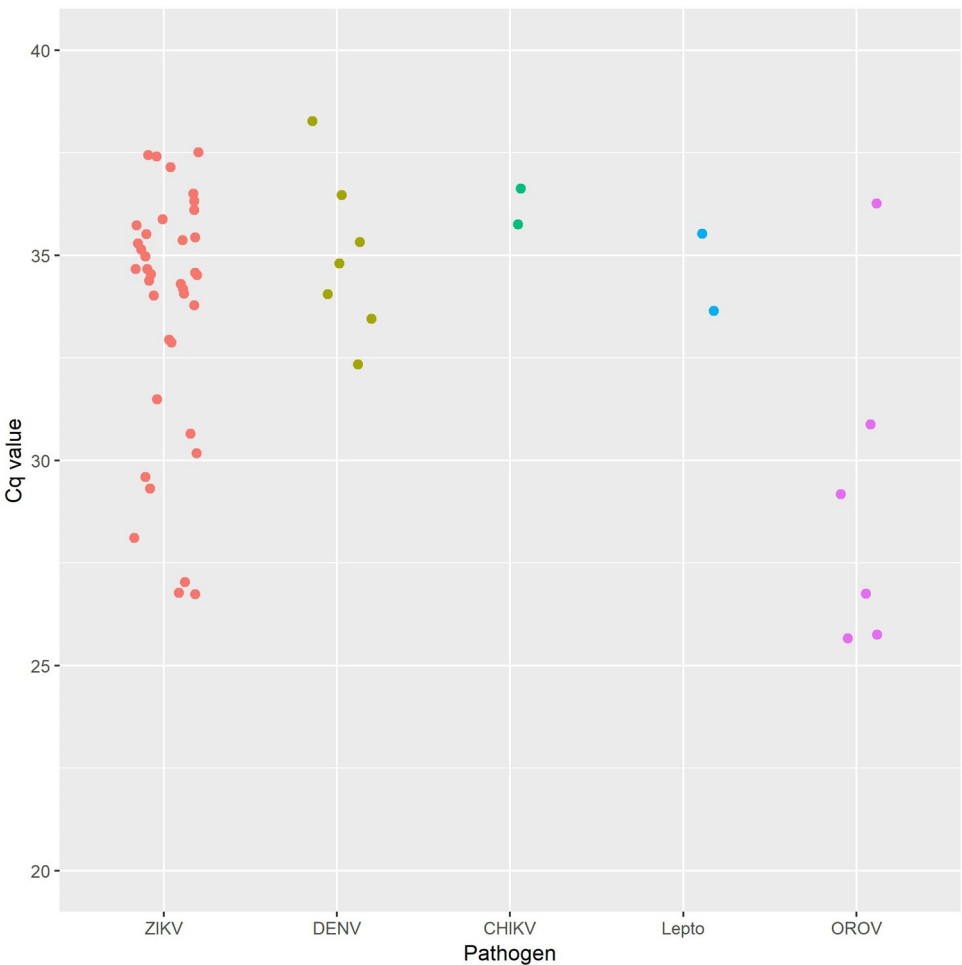

**Fig 2. Cq values from 49 pathogen positive or potentially positive plasma samples, from a cohort of 196 screened by qRT-PCR.** Samples with a Cq value of 35–39.9 should be considered potentially positive because (except for the OROV sample) confirmatory tests were not performed. Lepto = *Leptospira* spp.

## Pathogen confirmation and genome characterisation using metagenomic sequencing

All OROV qRT-PCR-positive samples were also OROV-positive by metagenomic sequencing, demonstrating that detection is possible from patient plasma samples with OROV RNA concentrations as low as 9.62e+3 genome copies/mL (S4 Table). Mapping reads to reference sequence OROV/EC/Esmeraldas/087/2016 resulted in genome coverage ranging from 68–99% (Fig 3). The proportion of OROV-specific reads does not appear to be particularly strongly correlated with Cq value, differences in the background level of non-viral RNA (in particular from the host) could account for this. It was still somewhat indicative of OROV genome coverage generated (Fig 3), with coverage generally decreasing as Cq increases. OROV reads were not detected in any of the 17 pathogen PCR-negative samples also tested. Hepatovirus A (HAV, formerly hepatitis A virus) was identified in one sample, with 1,894 reads providing 40% genome coverage (Table 1). Reads specific to internal control MS2 were identified in all PCR-negative samples.

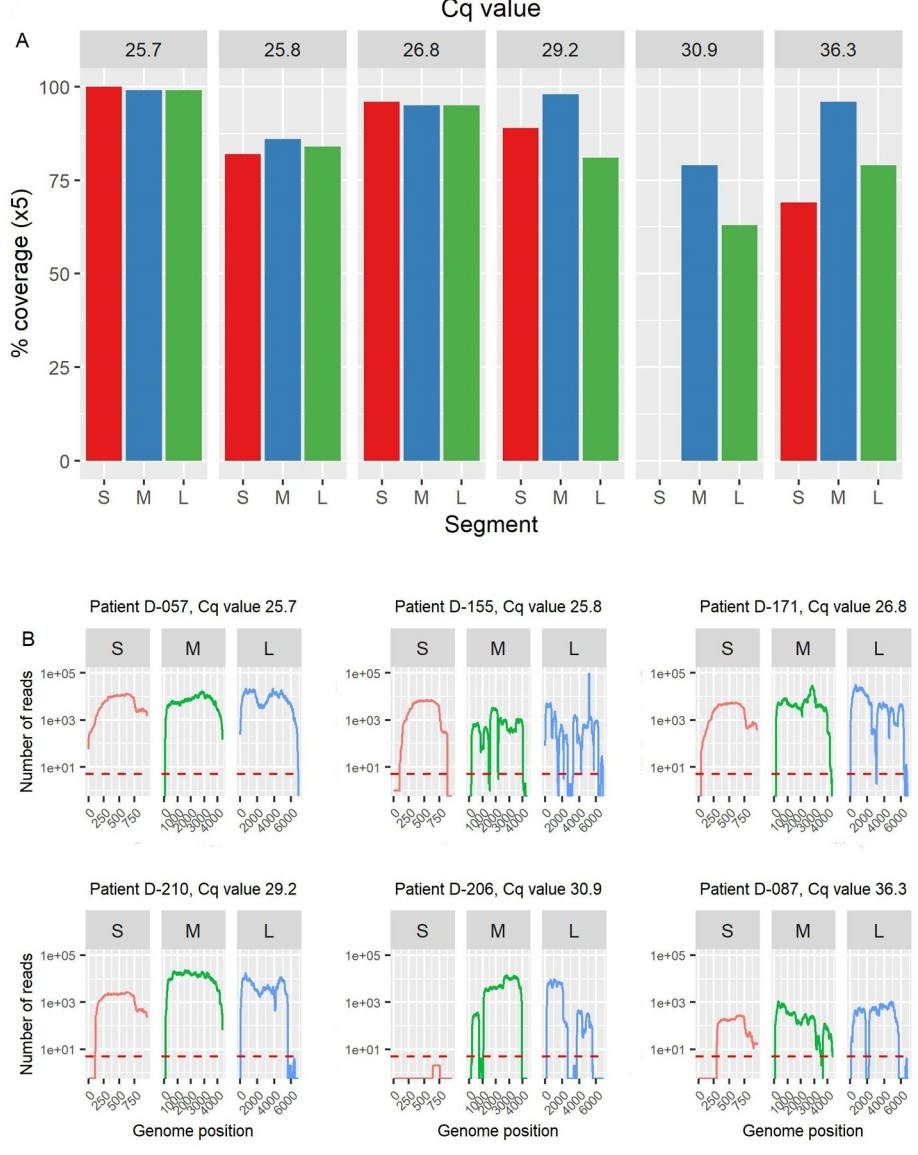

**Fig 3. OROV genome coverage generated from six OROV-positive patient plasma samples, using a metagenomic sequencing approach.** Reads were mapped to reference sequences OROV/EC/Esmeraldas/087/2016 (GenBank accessions MF926352.1- MF926354.1). **Panel A:** OROV genome segment (S, M, L) coverage (%). qRT-PCR Cq value is given at the top of each plot. Coverage is defined as a minimum of 5 reads at any given position. **Panel B:** OROV genome coverage shown as the number of reads at each genomic position. Plots are separated by genome segment (S, M, L). Red dashed line indicates 5x coverage.

## Confirmation of OROV infection in qRT-PCR-positive patients by virus isolation

OROV was successfully cultured in Vero cells from all five OROV-positive patient plasma samples (virus was isolated from the initial patient D-087 as previously reported [6]). S segment copy numbers for all samples increased by at least 5 logs to approximately 1.00e+12 copies/mL at 72 hours post-infection (S3 Fig), demonstrating that OROV was replicating. Confirmation of the presence of infectious virus was undertaken by plaque assay using P1 supernatant from the representative isolate OROV/EC/Esmeraldas/087/2016 from patient D-087, which formed plaques on Vero cells indicating a titre of 1.56e+7 pfu/mL. A subsequent

**Table 1. Viruses identified from 23 patient plasma samples by metagenomic sequencing, 17 of which were negative for all pathogens tested for by RT-PCR, including one sample (D-087) in which OROV was initially detecting by metagenomic sequencing.** The remaining five samples (D-057, D-155, D-171, D-206 and D-210) were initially shown to be positive for OROV using the qRT-PCR assay described in this study. The cut-off value used was ≥40% genome coverage (from mapping reads to reference genome).

| Sample ID | Virus identified | % of total reads assigned by Centrifuge | Reads assigned by Centrifuge | Reads mapped to reference sequence | 1x genome coverage (%) | Mapping reference (accession number) |
|---|---|---|---|---|---|---|
| D-005 | HAV | 0.02 | 478 | 1,894 | 40 | LC049339.1 |
| D-057 | OROV | 7.42 | 143,925 | 870,960 | 99 | MF926352.1–54.1 |
| D-087 | OROV | 0.70 | 5,084 | 9,972 | 80 | MF926352.1–54.1 |
| D-155* | OROV | 1.97 | 66,686 | 24,274 | 90 | MF926352.1–54.1 |
| D-171* | OROV | 3.13 | 94,248 | 145,146 | 95 | MF926352.1–54.1 |
| D-206 | OROV | 0.63 | 25,050 | 277,019 | 68 | MF926352.1–54.1 |
| D-210 | OROV | 4.15 | 78,747 | 733,986 | 92 | MF926352.1–54.1 |

*Data are from two separate sequencing runs combined.

passage (P2) on Vero cells, also assessed by plaque assay, further confirmed the presence of infectious virus.

RNA from cultured viral supernatant (one passage in Vero cells) underwent metagenomic sequencing and the resulting reads were mapped to reference genome OROV/EC/Esmeraldas/087/2016. Complete genomes were generated from all isolates and are available on GenBank, accession numbers MK506818—MK506832.

The genomes were 99.7–100% identical to one another in the S segment and 99.9–100% identical in the M and L segments. Single nucleotide polymorphisms (SNPs) were observed at 33 positions throughout the genome, all of which occurred in coding regions (S5 Table). Alignment of protein sequences showed that the N protein was 100% identical between isolates, whilst the NSs protein had a single amino acid substitution in isolate D-206. Isolate D-210 had two amino acid substitutions in the M protein, whilst three isolates (D-057, D-155 and D-206) had one substitution in this protein. Two isolates (D-155 and D-210) had a single amino acid substitution in the L protein (S6 Table). Of the eight substitutions seen, five constitute a reactive (R) group change (three on the M segment and two on the L segment) and therefore may influence the structure or function of the protein.

The genome sequences from the patient and cultured sequences were compared to identify SNPs incurred during passage. The number of SNPs per genome ranged from one (sample D-057) to 15 (sample D-155) (S7 Table). The majority of changes were from a mixed population (mixed defined as no one base occupying >80% of reads at a position) in the patient, to either a different mixed population, or a majority population (majority defined as one base occupying >80% reads at a position) in the cultured isolate (S8 Table). Changes from a majority population in the patient were less common, seen at two (samples D-171 and D-210) to five (sample D-087) positions, with sample D-057 showing no changes of this type (S8 Table). Only one of the 49 positions with polymorphisms between patient and cultured sequences was conserved between isolates (M segment position 59, S8 Table).

## Development of an OROV multiplex tiling RT-PCR primer scheme

Samples with lower levels of viral RNA often present challenges in generating complete genome sequences using the metagenomic sequencing approach (Table 1 and Fig 3). To address this problem, a set of targeted genome amplification primers was designed using the algorithmic method described by Quick *et al.* [16] (S2 Text). Using a cut-off of 20x read depth, >99% coverage of all three Ecuadorian OROV genome segments was obtained (average

coverage of 7,075x, 9,894x and 7,381x for the S, M and L segments, respectively) directly from clinical sample D-057 (Fig 4). For the prototype strain, 98% of the S segment (average coverage = 126,860x), 93% of the M segment (average coverage = 119x) and 92% of the L segment (average coverage = 189x) were obtained, despite its sequence divergence from the Ecuadorian strains.

## Discussion

Using metagenomic sequencing followed by screening using an optimised qRT-PCR, we have detected a cluster of six OROV cases from patients local to Esmeraldas, Ecuador, providing further evidence that OROV may be causing an unrecognised burden of human disease in previously unreported areas.

Some of these cases were not detected by the previously published assay due to a primer mismatch with the Ecuadorian genomes, which was identified in the metagenomic sequencing data. This discovery highlights the problems in developing amplification-based detection assays for pathogens with limited numbers of sequenced genomes, and the benefits offered by unbiased sequencing technologies. Although the cost associated with metagenomic sequencing may be prohibitive for frontline diagnosis in developing countries, it can be used in prospective screening studies to identify potential issues with qRT-PCR assays and allow for alterations to increase inclusivity, as reported here. We developed a new reverse primer for the previously published assay that improves sensitivity for both the Ecuadorian strain (capable of detecting 10 copies of target RNA) and the genetically divergent prototype strain. Other OROV strains were not available to validate against, however an alignment of all publicly available strains showed that the majority of sequences are homologous to the primer/probe sequences. The qRT-PCR amplifies a region in the S segment, the target sequence of which is shared by a number of reassortant orthobunyaviruses, including Iquitos virus, which is known to circulate in neighbouring Peru and can cause Oropouche fever [27]. This modified assay also has the potential to detect reassortant orthobunyaviruses containing the OROV S segment. An algorithmically-designed multiplex primer set was developed that can amplify the majority of the genome from both the Ecuadorian and prototype strains and may be a useful tool for sequencing OROV genomes from clinical samples with low viral titres.

Virus identification in the Ecuadorian cohort was undertaken initially using PCR. In addition to OROV, screening identified the presence of ZIKV, DENV, *Leptospira* spp. and possibly CHIKV nucleotide sequences. These pathogens are known to circulate in Ecuador [28–30]; ZIKV in particular was widespread in the Americas (including Ecuador) at the time of sampling [31], so the identification of a relatively large number of ZIKV cases ($n$ = 22 positive, $n$ = 15 potentially positive) was not unexpected. Cq values of 35–39.9 were observed for multiple samples across all identified pathogens. Samples with values in this range require confirmatory testing to be certain of the presence of the pathogen.

Metagenomic sequencing of 17 PCR-negative samples identified HAV sequence in one patient; no other viruses were conclusively identified. Ecuador is considered an 'intermediate endemicity' country for HAV [32] and therefore identification of this virus is not unexpected. The lack of viral sequences from the majority of the patient samples could be explained by a number of factors including clearance of virus from the blood by the time of sampling, or fever caused by non-infectious disease. Metagenomic sequencing successfully identified OROV in all OROV qRT-PCR-positive patients, with good genome coverage (68–99%), suggesting that this method is effective for both detection and characterisation of OROV genomes from clinical samples. Sequencing of one sample (D-206) produced M and L segment-specific reads that mapped to the reference sequence, however only two reads mapping to the S segment. The

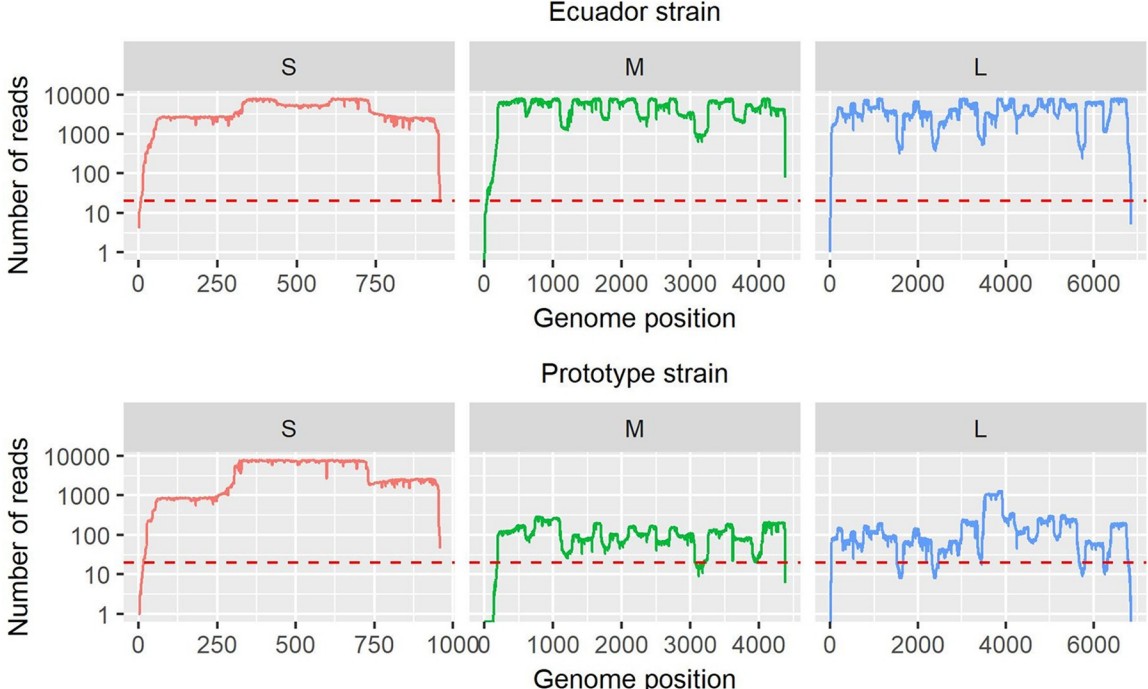

**Fig 4. Genome coverage of two OROV strains (Ecuadorian strain OROV/EC/Esmeraldas/087/2016 Genbank MF926352.1—MF926354.1, and prototype strain Genbank KP026179.1—KP026181.1), sequenced using a MinION (Oxford Nanopore), from multiplex tiling RT-PCR amplicons.** Plots are separated by genome segment (S, M, L). Red dashed line indicates 20x coverage.

absence of S segment signal here is simply due to the stochastic nature of the metagenomic sequencing method, which is exacerbated for smaller fragments. The S segment is both detected by qRT-PCR (at a Cq value of 30.9, the second highest value of all six OROV-positive patients) and is present in the sequencing of the virus directly cultured from this sample.

The generation of complete viral genomes using metagenomic sequencing or a multiplex tiling RT-PCR (as opposed to the small fragments generated using conventional qRT-PCR) enables more detailed and accurate molecular epidemiological analyses [33]. Initial phylogenetic analysis of S, M and L segments showed that the five additional Ecuadorian genomes cluster in a monophyletic group with our previously reported Ecuadorian strain, suggesting the possibility of local transmission within Esmeraldas. An investigation of potential OROV vector species in Esmeraldas would be highly informative. The group is most closely related to strain TVP-19256/IQE-7894 from Peru, 2008 (GenBank KP795084.1—KP795086.1). A comprehensive phylogenetic analysis of OROV in South America, including the Ecuadorian genomes, has been undertaken [26].

It is well-documented that mutations occur during passaging *in vitro* [34]. Our study generated complete genome sequences from viruses that had undergone one passage in Vero cells. The single passage was necessary to amplify virus and generate complete genome sequences, however, passaging was limited to minimise the number of mutations between patient and cultured viral genomes. A number of positions in the OROV patient genomes showed a mixed population of variants, which progressed to a majority population in the cultured genome. Majority-majority changes were rare and only one of the 49 SNPs observed between patient and cultured genomes was shared in two individual strains.

No cases of OROV were detected during follow-up qRT-PCR screening of 62 febrile patients from Esmeraldas, 2017. Previous outbreaks of Oropouche fever have been episodic

and self-limited in nature [4]; the six cases identified from 2016 could represent a single outbreak that ended that year. Alternatively, if OROV transmission did occur in Esmeraldas in 2017, the lack of detection could be related to the limited sample size of the cohort. A larger-scale study is necessary to determine the prevalence and distribution of OROV in the Ecuadorian population.

OROV infection should be considered when diagnosing Ecuadorian patients with febrile illness. We hope that the molecular tools developed in this study will be useful for laboratories performing viral diagnostics and surveillance in the Americas. Very little is known about the dynamics of OROV in Ecuador or endemicity in neighbouring regions; the detection of OROV in multiple febrile patients warrants further investigation into its prevalence, associated vectors, and transmission.

## Supporting information

**S1 Text. qRT-PCR assay optimisation and validation.**
(DOCX)

**S2 Text. Multiplex tiling PCR primer details.**
(DOCX)

**S1 Table. Oligonucleotides used in the development of the OROV qRT-PCR.**
(DOCX)

**S2 Table. Viruses tested for cross-reactivity with the qRT-PCR.**
(DOCX)

**S3 Table. Mismatches to oligonucleotide sequences observed in an alignment of 149 OROV N gene sequences.** Values are the number of sequences with mismatches to the primer/probe sequence.
(DOCX)

**S4 Table. Ecuadorian OROV-positive patient samples, determined by OROV S segment qRT-PCR.** Genome copies/ mL plasma are estimated based on the absolute quantitation standard curve. nd = no data.
(DOCX)

**S5 Table. SNPs identified between six Ecuadorian OROV genomes (sequenced from P1 Vero cell supernatant).** Variant base is shaded grey. * R position = 79% T, 21% C.
(DOCX)

**S6 Table. Amino acid variation between six Ecuadorian OROV genomes.** AA = amino acid. Gn = glycoprotein Gn. NSm = non-structural protein NSm. Gc = glycoprotein Gc. Bunyavirus Gn, NSm and Gc protein positions are taken from GenPept entry AGH07923.1. R group = reactive group.
(DOCX)

**S7 Table. A summary of the number of SNPs present in each OROV genome segment (S, M and L), between patient and cultured genome sequences.** n/a = no patient genome data available.
(DOCX)

**S8 Table. Positions in the OROV genome at which SNPs were identified between the patient and cultured genome, for each isolate.** Variance within each genome is also shown as the percentage of reads at that position showing a particular base. Seg. = segment. Seg. pos. =

segment position. Cons. = consensus.
(DOCX)

**S1 Fig. The workflow that led to the identification and isolation of OROV from six febrile Ecuadorian patients.**
(DOCX)

**S2 Fig. Absolute quantitation was performed from a standard curve generated from a ten-fold serial dilution of a synthetic OROV RNA standard.** Each data point is the mean Cq value from three separate experiments. Error bars indicate standard deviation. $R^2$ correlation coefficient = 0.9978.
(DOCX)

**S3 Fig. OROV genome copies increase over 96 hours in Vero cells, demonstrating OROV genome replication in five independent OROV cultures from OROV-positive patient plasma.**
(DOCX)

## Author Contributions

**Conceptualization:** Emma L. Wise, Simon K. Jackson, Gabriel Trueba, Gyorgy Fejer, Christopher H. Logue, Steven T. Pullan.

**Data curation:** Emma L. Wise, Bernardo Gutierrez, Oliver G. Pybus.

**Funding acquisition:** Simon K. Jackson, Gyorgy Fejer, Christopher H. Logue, Steven T. Pullan.

**Investigation:** Emma L. Wise, Sully Márquez, Jack Mellors, Verónica Paz.

**Methodology:** Emma L. Wise, Jack Mellors, Barry Atkinson.

**Resources:** Sully Márquez, Verónica Paz, Bernardo Gutierrez, Sonia Zapata, Josefina Coloma, Oliver G. Pybus.

**Supervision:** Simon K. Jackson, Gyorgy Fejer, Christopher H. Logue, Steven T. Pullan.

**Validation:** Emma L. Wise, Jack Mellors.

**Writing – original draft:** Emma L. Wise, Christopher H. Logue, Steven T. Pullan.

**Writing – review & editing:** Emma L. Wise, Sully Márquez, Jack Mellors, Verónica Paz, Barry Atkinson, Bernardo Gutierrez, Sonia Zapata, Josefina Coloma, Oliver G. Pybus, Simon K. Jackson, Gabriel Trueba, Gyorgy Fejer, Christopher H. Logue, Steven T. Pullan.

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
