## [Decision Letter · Decision Letter 0]

19 Sep 2019

Dear Miss Wise:

Thank you very much for submitting your manuscript "Oropouche virus cases identified in Ecuador using an optimised qRT-PCR informed by metagenomic sequencing" (PNTD-D-19-01189) for review by PLOS Neglected Tropical Diseases. Your manuscript was fully evaluated at the editorial level and by independent peer reviewers. The reviewers appreciated the attention to an important topic but identified some aspects of the manuscript that should be improved.

We therefore ask you to modify the manuscript according to the review recommendations before we can consider your manuscript for acceptance. Your revisions should address the specific points made by each reviewer.

(1) A letter containing a detailed list of your responses to the review comments and a description of the changes you have made in the manuscript.

(2) Two versions of the manuscript: one with either highlights or tracked changes denoting where the text has been changed (uploaded as a "Revised Article with Changes Highlighted" file); the other a clean version (uploaded as the article file).

(3) If available, a striking still image (a new image if one is available or an existing one from within your manuscript). If your manuscript is accepted for publication, this image may be featured on our website. Images should ideally be high resolution, eye-catching, single panel images; where one is available, please use 'add file' at the time of resubmission and select 'striking image' as the file type. 

Please provide a short caption, including credits, uploaded as a separate "Other" file. If your image is from someone other than yourself, please ensure that the artist has read and agreed to the terms and conditions of the Creative Commons Attribution License at http://journals.plos.org/plosntds/s/content-license (NOTE: we cannot publish copyrighted images). 

(4) Appropriate Figure Files 

Please remove all name and figure # text from your figure files upon submitting your revision. Please also take this time to check that your figures are of high resolution, which will improve both the editorial review process and help expedite your manuscript's publication should it be accepted. Please note that figures must have been originally created at 300dpi or higher. Do not manually increase the resolution of your files. For instructions on how to properly obtain high quality images, please review our Figure Guidelines, with examples at: http://journals.plos.org/plosntds/s/figures

While revising your submission, please upload your figure files to the Preflight Analysis and Conversion Engine (PACE) digital diagnostic tool, https://pacev2.apexcovantage.com/ PACE helps ensure that figures meet PLOS requirements. To use PACE, you must first register as a user. Then, login and navigate to the UPLOAD tab, where you will find detailed instructions on how to use the tool. If you encounter any issues or have any questions when using PACE, please email us at figures@plos.org.

We hope to receive your revised manuscript by Nov 18 2019 11:59PM. If you anticipate any delay in its return, we ask that you let us know the expected resubmission date by replying to this email.

To submit your revised files, please log in to https://www.editorialmanager.com/pntd/

Sincerely,

Mariangela Bonizzoni

Associate Editor

A. Desiree LaBeaud

Deputy Editor

Reviewer's Responses to Questions

**Key Review Criteria Required for Acceptance?**

**Methods**

-Are the objectives of the study clearly articulated with a clear testable hypothesis stated?

-Is the study design appropriate to address the stated objectives?

-Is the population clearly described and appropriate for the hypothesis being tested?

-Is the sample size sufficient to ensure adequate power to address the hypothesis being tested?

-Were correct statistical analysis used to support conclusions?

-Are there concerns about ethical or regulatory requirements being met?

Reviewer #1: (No Response)

Reviewer #2: All methods are adequate for the hypothesis to be tested, the sample size is sufficient and the statistical analysis support well the conclusions.

Reviewer #3: The objectives of the study are well defined and clearly articulated. The methodology is well written, and the large sample size used here is a strong point of this manuscript. There are some areas which should be expanded with greater details given, on order to aid the reader.

Specific comments:

• RNA extraction and PCR assays: Are the in house PHE assays published? If so, they should be referenced here.

• LOD analysis: what was the rationale behind using a 711bp fragment of S segment, rather than the entire S genome for the in vitro transcript control material?

**Results**

-Does the analysis presented match the analysis plan?

-Are the results clearly and completely presented?

-Are the figures (Tables, Images) of sufficient quality for clarity?

Reviewer #1: (No Response)

Reviewer #2: The results presented here matched well the analysis plan. All results were clear and well presented. The figures and all supporting material are of good quality.

Reviewer #3: The results section is for the most part clearly written, easy to follow, and the data are presented in a logical and methodical way. I think it would be beneficial to include more details in all the figure and table legends as they are rather sparse - such as (where appropriate): GenBank numbers, sample IDs, number of samples per group.

One section I think could be improved on is the confirmation of infectious OROV in the positive plasma. Here, the authors describe a method where they monitored S genome levels over time in vitro, and used increasing viral RNA levels as proof of viral isolation. While this observation may be true, I feel this would be a much more convincing statement if there was more direct evidence that OROV was actively replicating and producing new virions in vitro. Ultimately, a P2 passage onto Vero cell would be the gold standard here, but evidence of CPE (OROV in Vero cells causes extensive cell death) or protein production (e.g. IFA analysis) should be reported. One reason I think it’s very important to confirm active and productive viral replication is that the S genome copies/ml determined by qRT-PCR reported here are very high. At this level (even though it is difficult to equate RNA copy numbers to actual virion numbers) I would think the titer would be capable of destroying a Vero monolayer in ~24 hours, whereas data presented here would suggest at least some cell survival up to 96 hpi?

Specific comments:

• Development: There are several “reduced sensitivity” and “improved sensitivity” statements in this section, and some are quantified (i.e. 3 logs) and some are not. If possible, all variations in sensitivity should be consistently quantified by some measure.

• Development: How many publicly available OROV N gene sequences are there (n=?). This should be stated.

• Fig 2: These data should be performed/presented in at least triplicate, and analyzed using a trend line to determine efficiency and sensitivity difference between both viral strains and assays (similar to analysis in Fig S1).

• Was there a specific reason why plasma samples from 2016 and 2017 were sampled separately? If so, this should be stated in the text.

• Confirmation of infectious OROV: Should say “S genome copies” rather than “genome copies”. The number of reactive group substitutions (of the 5 total) on each segment should be stated.

• Supplementary Table 6: I think it would be useful to the reader to know what these SNPs were rather than just the number per segment.

**Conclusions**

-Are the conclusions supported by the data presented?

-Are the limitations of analysis clearly described?

-Do the authors discuss how these data can be helpful to advance our understanding of the topic under study?

-Is public health relevance addressed?

Reviewer #1: (No Response)

Reviewer #2: The conclusion support well the data presented and I did not see any limitation of the study. All data are well discussed and the paper is a good source of information about the history of the Oropouche virus. This paper is a great public relevance as it highlights the problems of OROV in Ecuador and Latin America.

Reviewer #3: The discussion is well written, and both expands on the results and brings in valid discussion points as to the relevance of these data.

One specific comment that isn’t explained in the results, and perhaps should be touched upon in the discussion in sample D-206. This sample was PCR positive, had a large number of reads mapping to the reference sequence, but yet no S genome coverage was obtainable using the NGS? Do the authors have any explanation for this? NGS results like this would suggest a reassortant (although an S-only bunyavirus reassortants would be very rare), but the qRT-PCR seems to contradict this. Even if ultimately this result was due to an assay failure, this should at least be brought up in the discussion rather than left unexplained.

**Editorial and Data Presentation Modifications?**

Reviewer #1: (No Response)

Reviewer #2: I do not have major editorial suggestions, only to encourage the authors to double check all references and format of the journal.

Reviewer #3: (No Response)

**Summary and General Comments**

Reviewer #1: In this study the authors develop a new diagnostic tool for Oropouche virus, a neglected arbovirus circulating in South America. The optimization and validation of qRT-PCR was performed basing on a metagenomic data obtained by sequencing human plasma sample from Ecuador. The presented qRT-PCR was able to identify OROV in samples previously tested as negative. The results presented will improve the capacity to detect of OROV in endemic regions. The paper is well written, and all data are available. All my concerns are minor.

1) I would like to read some more information in the introduction about the vectors involve in the transmission cycle of OROV, also considering that the authors highlight the need of more vector competence studies in discussion. 

2) Figure 1 does not seem to add much to the paper as it’s a technical type figure. i suggest to delete it.

3) Lines 100-105: in this chapter of MM the authors don’t report the Genbank number of the sequence used. I try to find them in the cited paper but I was not able to find the data. I suggest to clarify this point.

4) Considering the main object of this paper, I would like to see the primer sequences in the main text and not only in a table in supplementary information. 

5) Lines 313-318: the authors have report positive Cq values of 35-40 (table 3) but these data require confirmatory test. If they don’t provide the confirmation of these results I would be conservative and indicate the samples as possible positive.

Reviewer #2: I really enjoyed reading the paper and the methodologies used to detect OROV in Ecuador. I congratulate the authors in devising a new set of primers that detect variants of OROV.

I encourage the publication of this paper.

Reviewer #3: In the manuscript “Oropouche virus cases identified in Ecuador using an optimized qRT-PCR informed by metagenomic sequencing”, Wise et al. describe the use of metagenomic sequencing to aid in the design of PCR assays to detect OROV, and perform analysis on historical samples from patients with undiagnosed fevers to investigate OROV in an endemic area. The experiments are well thought out, the paper is well written, and the data presented in a clear and concise manner. 

My one suggestion to increase the strength of this manuscript would be to perform further work on the experiments investigating OROV isolation from the patient plasma samples. This data would benefit from extra experimental work to confirm presence of replicating OROV.

PLOS authors have the option to publish the peer review history of their article (what does this mean?). If published, this will include your full peer review and any attached files.

Reviewer #1: No

Reviewer #2: No

Reviewer #3: No

---

## [Editor Report · Decision Letter 1]

31 Oct 2019

Dear Miss Wise,

We are pleased to inform you that your manuscript, "Oropouche virus cases identified in Ecuador using an optimised qRT-PCR informed by metagenomic sequencing", has been editorially accepted for publication at PLOS Neglected Tropical Diseases.

Before your manuscript can be formally accepted and sent to production you will need to complete our formatting changes, which you will receive in a follow up email. Please note: your manuscript will not be scheduled for publication until you have made the required changes.

IMPORTANT NOTES

* Copyediting and Author Proofs: To ensure prompt publication, your manuscript will NOT be subject to detailed copyediting and you will NOT receive a typeset proof for review. The corresponding author will have one final opportunity to correct any errors when sent the requests mentioned above. Please review this version of your manuscript for any errors.

* If you or your institution will be preparing press materials for this manuscript, please inform our press team in advance at plosntds@plos.org. If you need to know your paper's publication date for media purposes, you must coordinate with our press team, and your manuscript will remain under a strict press embargo until the publication date and time. PLOS NTDs may choose to issue a press release for your article. If there is anything that the journal should know, please get in touch.

*Now that your manuscript has been provisionally accepted, please log into EM and update your profile. Go to http://www.editorialmanager.com/pntd, log in, and click on the "Update My Information" link at the top of the page. Please update your user information to ensure an efficient production and billing process.

*Note to LaTeX users only - Our staff will ask you to upload a TEX file in addition to the PDF before the paper can be sent to typesetting, so please carefully review our Latex Guidelines [http://www.plosntds.org/static/latexGuidelines.action] in the meantime.

Best regards,

Mariangela Bonizzoni

Associate Editor

A. Desiree LaBeaud

Deputy Editor

---

## [Editor Report · Acceptance letter]

9 Jan 2020

Dear Miss Wise,

We are delighted to inform you that your manuscript, "Oropouche virus cases identified in Ecuador using an optimised qRT-PCR informed by metagenomic sequencing," has been formally accepted for publication in PLOS Neglected Tropical Diseases.

Best regards,

Serap Aksoy

Editor-in-Chief

Shaden Kamhawi

Editor-in-Chief
